# Enantioselective Ugi and Ugi-azide reactions catalyzed by anionic stereogenic-at-cobalt(III) complexes

Bing-Bing Sun[1,3], Kun Liu[1,3], Quan Gao[2,3], Wei Fang[1], Shuang Lu[1], Chun-Ru Wang[1], Chuan-Zhi Yao[1], Hai-Qun Cao [2] ✉ & Jie Yu [1,2] ✉

Ugi reactions and related variations are proven to be atom and step-economic strategies for construction of highly valuable peptide-like skeletons and nitrogenous heterocycles. The development of structurally diverse range of novel catalytic systems and the discovery of new approaches to accommodate a broader scope of terminating reagents for asymmetric Ugi four-component reaction is still in high demand. Here, we report a strategy that enables enantioselective Ugi four-component and Ugi-azide reactions employing anionic stereogenic-at-cobalt(III) complexes as catalysts. The key nitrilium intermediates, generated through the nucleophilic addition of isocyanides to the chiral ion-pair which consists of stereogenic-at-cobalt(III) complexes counteranion and a protonated iminium, are trapped by either carboxylic acids or in situ-generated hydrazoic acid, delivering α-acylamino amides and α-aminotetrazoles in good to excellent enantioselectivities (up to 99:1 e.r.).

Multicomponent reactions (MCRs) can assemble three or more starting materials in a single operation to rapidly build up molecular complexity and diversity, providing invaluable access to bioactive molecules with significant atom- and step-efficiency[1-5]. A premier example is the Ugi-MCR[6] that transforms amines, carbonyl compounds (aldehydes or ketones), isocyanides, and appropriate heteronucleophiles to α-acylamino amides and heterocyclic amines or amides[7-9], which are the core skeletons of a variety of compounds with medicinal value, such as Ivosidenib, potent p53-MDM2 antagonist, HCV NS5B polymerase, and BMS-317180 (Fig. 1a)[10,11]. Mechanistically, the nucleophilic α-addition of isocyanides to in situ-generated imines furnish the key nitrilium intermediates, which can react with diverse terminating reagents to afford those intriguing structural patterns[12-15].

The nitrilium ion trapping in classical Ugi four-component reaction (Ugi-4CR) falls on carboxylic acids, which are generally believed to play a dual role: (1) impart LUMO activation on imines to facilitate the addition of isocyanides; (2) serve as terminating reagents to access the peptide-like moiety from nitrilium species and subsequent Mumm rearrangement[16-18]. The abundance of carboxylic acids, along with

amines and carbonyl compounds, has enabled the practical synthesis of heterocyclic small molecules[12-15] and macromolecules[19-21]. Despite these great advances, stereochemical control remains the most significant challenge still facing Ugi-4CR. To achieve catalytic Ugi-4CR with excellent enantioselectivity, several limitations need to be properly addressed: (1) uncatalyzed background reaction affording racemic products; (2) catalytic modes dysfunction due to the striking similarity of most functionalities within the reaction system; (3) difficult stereocontrol in the α-addition of isocyanides to imines. Great efforts have been devoted to the discovery of high-performance catalysts, which indeed offer a solution for these challenges[22,23].

In simpler Ugi three-component reaction (Ugi-3CR) systems[24], asymmetric organocatalysis has been proven to be robust[25]. A significant breakthrough came in 2009 that Wang, Zhu, and coworkers employed in situ-generated chiral phosphoric acid-protonated iminium salts for the addition of α-isocyanoacetamides, furnishing chiral 5-aminooxazoles in one step[26]. Later, this chiral phosphoric acid catalytic strategy was expanded to the Ugi four-center, three-component reaction of isocyanides, anilines, and 2-formyl benzoic acid[27,28].

[1]Department of Applied Chemistry, Anhui Agricultural University, Hefei 230036, China. [2]School of Plant Protection, Anhui Province Engineering Laboratory for Green Pesticide Development and Application, and Anhui Province Key Laboratory of Crop Integrated Pest Management, Anhui Agricultural University, Hefei 230036, China. [3]These authors contributed equally: Bing-Bing Sun, Kun Liu, Quan Gao. ✉e-mail: haiquncao@163.com; jieyu@ustc.edu.cn

Maruoka et al. developed chiral dicarboxylic acids for an enantioselective Ugi-type reaction of acyclic azomethine imines and isocyanides[29] (Fig. 1b). Wulff and coworkers disclosed a chiral boroxinate anion involved Ugi-3CR of dibenzylamine, aromatic aldehydes, and isocyanides[30]. The enantioselective Ugi-4CR of amines, aldehydes, isocyanides, and carboxylic acids was unconquered[22] until 2018 by the milestone work of Tan, Houk, and coworkers[31,32]. It was proposed that a hydrogen-bonding-network-stabilized complex, consisted of a chiral phosphoric acid, an imine, and a carboxylic acid, directed the nucleophilic addition of the isocyanide to the hydrogen-bonding activated imine, which was identified as the key of success in achieving excellent enantioselectivity (Fig. 1c). Despite these advances, the development of novel catalysts to accommodate broader scope of terminating reagents for asymmetric Ugi-4CR and related variations[33] is still in high demand[34].

Distinct from the traditional chiral metal complexes, octahedral stereogenic-at-metal complexes have emerged as privileged scaffolds in asymmetric catalysis[35–38]. Anionic stereogenic-at-cobalt(III) complexes in which the center metal Co(III) was coordinatively-saturated, were turned out to be excellent catalysts for asymmetric Povarov reaction of 2-azadienes in high stereoselectivities, wherein the weakly-coordinating nature of chiral ion-pair was proposed to permit the alkali cation to work as a Lewis acid for the activation of imine functionality[39]. Recently, anionic stereogenic-at-cobalt(III) complexes were also introduced as either anionic catalysts[40–42] or weakly-coordinating anions[43], capable of catalyzing the enantioselective bromocyclization of olefins[44,45], C(sp³)-H arylation of thioamides[46,47], and atroposelective ring-opening reaction of cyclic diaryliodonium salts with bulky anilines[48] (Fig. 1d). These examples showcased the robustness of anionic stereogenic-at-cobalt(III) complexes catalysis, which, in our envision, would continue to improve the stereocontrol in practical yet challenging catalytic asymmetric synthesis.

In this work, we demonstrate the anionic stereogenic-at-cobalt(III) complexes catalysis strategy for enantioselective Ugi-MCRs, including Ugi four-component and Ugi-azide reactions. The key step of stereocontrol is realized via the nucleophilic addition of isocyanide to the chiral ion-pair which consists of stereogenic-at-cobalt(III) complexes counteranion and a protonated iminium (Fig. 1e)[16–18]. Interestingly,

through modulating terminating reagents, the resulting transient nitrilium intermediates are trapped by either carboxylate or azide, leading to enantioenriched Mumm rearrangement products and tetrazole derivatives, respectively, in good to excellent enantioselectivities.

## Results

### Optimization of reaction conditions

Our initial attempts commenced with the Ugi-4CR by using 4-bromobenzaldehyde **2a**, 4-(trifluoromethyl)aniline **3a**, tert-butyl isocyanide **4a**, and benzoic acid **5a** as reactants and stereogenic-at-cobalt(III) complex-templated Brønsted acid Λ-**1a** as a catalyst in toluene at −20 °C with 4 Å molecular sieves (see Supplementary Table 1 for details). To our delight, the reaction underwent smoothly to delivered the desired Ugi-4CR product **6** in 80.5:19.5 e.r. (Table 1, entry 1). Either Brønsted acids or sodium salts of anionic stereogenic-at-cobalt(III) complexes were then screened in the reaction (entries 2-12) and it was found that sodium salt Λ-**1h** which derived from 3,5-di-tert-butyl-substituted salicylaldehyde and L-tert-leucine gave the highest enantiomeric ratio of 95:5 among them (entry 8). Comparisons between lithium salt **1m**, potassium salt **1n** and sodium salt **1h** suggested that the type of the cations has little influence on the enantioselectivity (entries 13-14 vs entry 8). The introducing of a larger substituent (such as trimethylsilyl, or triethylsilyl) at the C3 position (R″) of the salicylaldehyde moiety[39] did not further improve the reaction outcome (entries 15-16). Δ-(S,S)-**1b**, the diastereomer of Λ-(S,S)-**1b**[44], enabled the reaction to give the *ent*-**6** in 72:28 e.r. (entry 17). After fine-tuning of the ratio of the four components and screening of solvents, the best result with 99% yield and 97.5:2.5 e.r. was achieved when Ugi-4CR was carried out in toluene at −40 °C for 12 h (entry 18 vs entries 19-22).

To our delight, the use of terminating reagent TMSN₃ to replace benzoic acid **5a** provide the Ugi-azide reaction product **7** in 76% yield with enantiomeric ratio of 94.5:5.5 (entry 23, see Supplementary Table 2 for details). Other azides, such as TsN₃ or NaN₃, were ineffective for this reaction, in which a trace amount of adduct **7** was afforded (entries 24-25). Interestingly, introducing methylamine hydrochloride as a co-additive could deliver α-aminotetrazole **7** in 79% yield and 93:7 e.r. (entry 26), which implies that hydrazoic acid might be generated

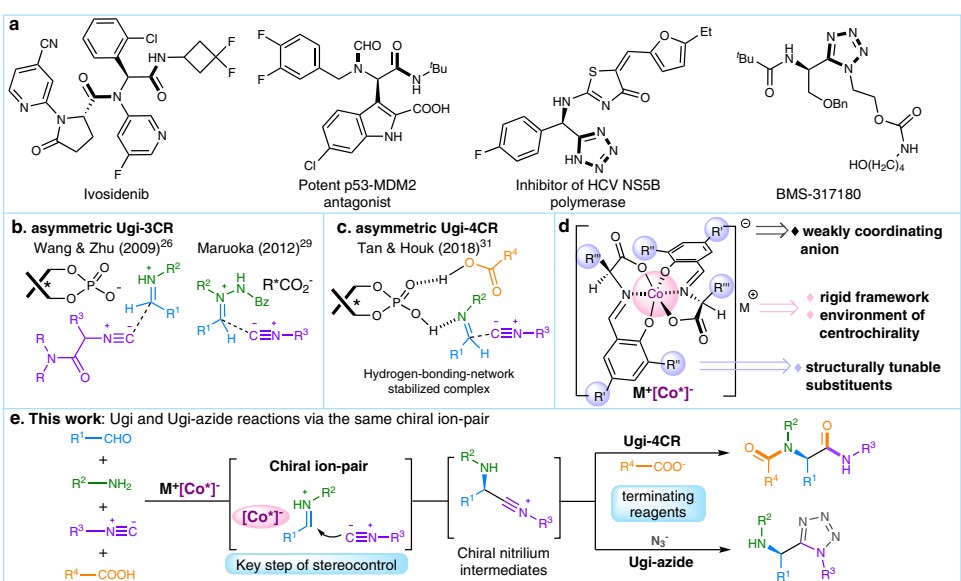

**Fig. 1 | Background of asymmetric Ugi-type reactions and our strategy.** **a** Selected bioactive molecules prepared from Ugi-MCRs. **b** Proposed stereocenter-inducing complex in the representative catalytic Ugi-3CR. **c** Schematic drawing of the stereocenter-inducing complex in the chiral phosphoric acid-catalyzed Ugi-

4CR. **d** The feature of anionic stereogenic-at-cobalt(III) complexes. **e** Our strategy: anionic stereogenic-at-cobalt(III) complexes as efficient catalysts for asymmetric Ugi and Ugi-azide reactions (this work).

in situ and the formation of protonated iminium under the acidic conditions should be beneficial to the nucleophilic addition of isocyanide. The enantiomeric ratio of **7** increased to 95:5 when acetic acid **5b** was used instead of methylamine hydrochloride (entry 27), while a little amount of Ugi-4CR product (11% yield) was also formed. Modulating the equivalent of **5b** and the reaction temperature could improve the yield to 82% and the enantioselectivity to 95.5:4.5 e.r. (entry 28).

### Reaction scope of enantioselective Ugi four-component reactions

With the optimal conditions in hand (Table 1, entry 18), the generality of Ugi-4CRs was first examined (Fig. 2). The enantioselective Ugi 4CR catalyzed by anionic stereogenic-at-cobalt(III) complexes could be scaled up. Even the presence of 5 mol% Λ-**1h** was able to render the gram-scale reaction to give **6** in 92% yield with 97:3 e.r. Aside from 4-(trifluoromethyl)aniline **3a**, a variety of anilines **3** with either electron-withdrawing or halogen substituents could be converted to the α-acylamino amides **8-14** with good to excellent isolated yields (52-99%) and enantioselectivities (91:9 to 97.5:2.5 e.r.). A broad range of the acid components was well tolerated under mild conditions, as products bearing aromatic carboxylic acids (**15-17**), cinnamic acid (**18**), linear alkyl carboxylic acids (**19, 23-26**), cycloalkyl carboxylic acids (**20-22**), bulky pivalic acid (**27**), 2-furoic acid (**28**) and sorbic acid (**29**) could be obtained with good enantioselectivities (up to 97:3 e.r.).

Furthermore, the substrate scope was successfully extended to the aldehydes **2**. Aromatic aldehydes with varied functionalities were able to undergo the reactions smoothly to provided expected Ugi products **30-42** in excellent yields with enantioselectivities (up to 97:3 e.r.). Both the electronic property and the position of the substituents on the aromatic ring did not exert an obvious effect on the reactivity and enantioselectivity of the process. Only the participation of the aldehydes which bear either *ortho*-substituted groups (**30-31**) or *para*-substituted electron-donating groups (**40**) led to a slightly diminished enantioselectivities. The absolute stereochemistry of **31** was determined to be R-configuration by single-crystal X-ray diffraction analysis (CCDC no. 2103289), and those of the others were assigned analogously. The reaction of β-naphthaldehyde and furfural afforded Ugi products **43** and **44** with 95:5 and 92.5:7.5 e.r., respectively. The use of 4-hydroxybenzaldehyde, which contains an acidic functionality, as the aldehyde component led to a significantly diminished enantioselectivity (**45**). The Ugi-4CRs did not occur in toluene at −40 °C when aliphatic aldehydes were used. Therefore, a series of other solvents, such as CH₂Cl₂, CHCl₃, and methanol, were then screened as co-solvents. Fortunately, when the reactions of cyclohexanecarbaldehyde with different anilines were carried out in a toluene/methanol mixture (20:1 v/v), α-acylamino amides **46-48** could be obtained with enantiomeric ratios from 81:19 to 91.5:8.5. The aliphatic α-unbranched aldehydes underwent Ugi 4CRs with 4-aminobenzonitrile smoothly to generate the desired products **49-51** with good enantioselectivities. However, the introduction of butylamine led to a substantial decrease in enantioselectivity (**52**). Although ethyl isocyanoacetate formed **58** with moderate optical purity, primary, secondary, and tertiary alkyl isonitriles formed Ugi products (**53−57, 59**) in excellent enantioselectivities (up to 98:2 e.r.), regardless of the effect of steric hindrance.

### Reaction scope of enantioselective Ugi-azide reactions

Having established optimal conditions for Ugi-azide reactions (Table 1, entry 28), the substrate scope of this protocol was evaluated (Fig. 3). Under the optimal reaction conditions, 1.790 g of **7** could be obtained in 79% yield and 94:6 e.r. in the presence of 5 mol% Λ−**1h**, highlighting the practicality of this transformation. Absolute configuration of **7** was determined as R by X-ray diffraction analysis (CCDC no. 2103292). Employing Λ-**1h** as the same catalyst of asymmetric Ugi-4CR, broadly good to excellent outcomes (up to 99:1 e.r.) were extendable to

aromatic aldehydes with various substituents (**60-74**). The level of enantioselectivity in tetrazole **75** derived from furfural in Ugi-azide reaction was identical to that generated in Ugi-4CR. β-Naphthaldehyde could give rise to **76** in 59% yield with 88.5:11.5 e.r. Tetrazoles **77** and **78** formed from 2-pyridyl carboxaldehyde and 4-hydroxybenzaldehyde were obtained with moderate enantioselectivity, respectively. Similar to the conditions of Ugi-4CRs for aliphatic aldehydes, the reactions of either an α-branched aldehyde or an α-unbranched aldehyde were able to give the α-aminotetrazoles **79-80** with up to 87.5:12.5 e.r. when methanol was used as a co-solvent. A series of anilines were then investigated using 4-cyanobenzaldehyde as the aldehyde component. It turned out that different anilines were able to give tetrazoles **81-90** with up to 98:2 e.r. and the electronic properties of the aromatic ring do not seem to influence the enantioselectivity of the reaction. The tolerance of isocyanides in Ugi-azide reactions was next examined. Compared with aliphatic isocyanides (**91-95, 98**), the reaction of α-isocyanoacetate could deliver Ugi-azide products **96-97** in medium yield (46-49%) and excellent enantioselectivity (up to 97:3 e.r.). Moreover, selected trifluoromethyl group-containing α-acylamino amides and tetrazoles were tested against three phytopathogenic fungi (*Colletotrichum gloeosporioides* Penz., *Botrytis cinerea* and *Fusarium oxysporum* (Schl.) F. sp. *cucumerinum* Owen), which have shown potential antifungal activities and can be regarded as promising candidates in the search for new pesticide scaffolds (see Supplementary Table 3 & 4 for details).

### Control experiments and mechanistic studies

In order to get insight into the reaction mechanism, we conducted several control experiments. Either Ugi or Ugi-azide reaction with preformed aldimine **99a** was successful, furnishing the products with enantioselectivities similar to those observed for the corresponding reactions with aldehydes and anilines, indicating that trans-aldimine has been formed before the catalytic cycle (Fig. 4a).

Different from Tan's work[31], these reactions can be carried out in the presence of two equivalents of aniline **3a**, delivering the corresponding products albeit with slight decreases in enantioselectivity (Fig. 4b). Interestingly, the Ugi-3CR didn't work in the absence of acid component, whereas the introduction of hydrochlorides or methanesulfonic acid as co-additives could lead the formation of Ugi-3CR product **100** with up to 71:29 e.r. (Fig. 4c). It is suggested that the imine might be activated by a proton transfer in the presence of excess carboxylic acids, leading to iminium intermediate, rather than hydrogen-bonding activated imine[49,50].

During the studies on the catalytic Ugi-4CRs, we found a positive nonlinear effect for the reactions of aldehyde **2a**, aniline **3a**, isocyanide **4a**, with either benzoic acid **5a** or acetic acid **5b** in the presence of 10 mol % of the scalemic catalyst **1h** (see Supplementary Table 8 & 9 for details). A slight positive nonlinear effect was also observed for the Ugi-azide reaction of aldehyde **2a**, aniline **3a**, isocyanide **4a**, acetic acid **5b** with NaN₃, suggesting that only one catalyst molecule might be involved in the enantio-determining transition states (Fig. 5a). As depicted in Table 1, Ugi-4CR product **23** was generated when the asymmetric Ugi-azide reaction was conducted in the presence of NaN₃ and acetic acid. Modulation of the ratio of NaN₃/acetic acid did not avoid the formation of **23**. The enantioselectivity of **7** could drop from 95.5:4.5 to 91:9 e.r. when NaN₃ loading was increasing from 3 equiv. to 7 equiv. (Fig. 5b, see Supplementary Table 5 for details).

A series of carboxylic acids with different p$K_a$ and steric properties led to the Ugi-4CR products **15-29** with enantioselectivities ranging from 79:21 to 97:3 e.r. Aside from acetic acid **5b**, the Ugi-azide reactions with various carboxylic acids were also examined, and it was found that acetic acid **5b** gave the best result. Both the steric and electron nature of the carboxylic acids had a significant effect on the asymmetric introduction (see Supplementary Table 2 for details). It is revealed that the carboxylic acids did participate in the key step of

**Table 1 | Optimization studies for asymmetric Ugi and Ugi-azide reactions**

1a: R' = R'' = $^t$Bu, R''' = $^i$Pr, M = H;
1b: R' = R'' = $^t$Bu, R''' = $^i$Pr, M = Na;
1c: R' = R'' = $^t$Amyl, R''' = $^i$Pr, M = H;
1d: R' = R'' = $^t$Amyl, R''' = $^i$Pr, M = Na;
1e: R' = R'' = $^t$Amyl, R''' = $^i$Pr, M = Na;
1f: R' = R'' = $^t$Amyl, R''' = $^t$Bu, M = H;
1g: R' = R'' = $^t$Bu, R''' = $^t$Bu, M = H;
1h: R' = R'' = $^t$Bu, R''' = $^t$Bu, M = Na;
1i: R' = R'' = $^t$Bu, R''' = Bn, M = Na;
1j: R' = R'' = $^t$Bu, R''' = $^t$Bu, M = Na;
1k: R' = R'' = $^t$Bu, R''' = $^s$Bu, M = Na;
1l: R' = R'' = $^t$Bu, R''' = Cy, M = Na;
1m: R' = R'' = R''' = $^t$Bu, M = Li;
1n: R' = R'' = R''' = $^t$Bu, M = K;
1o: R' = R'' = $^t$Bu, R''' = TMS, M = Na;
1p: R' = R'' = $^t$Bu, R''' = TES, M = Na.

| Entry | 1 | solvent | terminating reagents | 6 yield (%)[a] | 6 e.r.[b] | 7 yield (%)[a] | 7 e.r.[b] |
|---|---|---|---|---|---|---|---|
| 1 | Λ-(S,S)-1a | toluene | 5a | 54 | 80.5:19.5 | - | - |
| 2 | Λ-(S,S)-1b | toluene | 5a | 33 | 85.5:14.5 | - | - |
| 3 | Λ-(S,S)-1c | toluene | 5a | 63 | 80.5:19.5 | - | - |
| 4 | Λ-(S,S)-1d | toluene | 5a | 61 | 84.5:14.5 | - | - |
| 5 | Λ-(S,S)-1e | toluene | 5a | 77 | 88.5:11.5 | - | - |
| 6 | Λ-(S,S)-1f | toluene | 5a | 64 | 93:7 | - | - |
| 7 | Λ-(S,S)-1g | toluene | 5a | 89 | 92:8 | - | - |
| 8 | Λ-(S,S)-1h | toluene | 5a | 72 | 95:5 | - | - |
| 9 | Λ-(S,S)-1i | toluene | 5a | trace | N.D. | - | - |
| 10 | Λ-(S,S)-1j | toluene | 5a | 27 | 71:29 | - | - |
| 11 | Λ-(S,S)-1k | toluene | 5a | 18 | 69:31 | - | - |
| 12 | Λ-(S,S)-1l | toluene | 5a | 39 | 65.5:34.5 | - | - |
| 13 | Λ-(S,S)-1m | toluene | 5a | 70 | 94.5:5.5 | - | - |
| 14 | Λ-(S,S)-1n | toluene | 5a | 83 | 93.5:6.5 | - | - |
| 15 | Λ-(S,S)-1o | toluene | 5a | 96 | 93:7 | - | - |
| 16 | Λ-(S,S)-1p | toluene | 5a | 40 | 95:5 | - | - |
| 17 | Δ-(S,S)-1b | toluene | 5a | 39 | 28:72 | - | - |
| 18[c] | Λ-(S,S)-1h | toluene | 5a | 99 | 97.5:2.5 | - | - |
| 19[c] | Λ-(S,S)-1h | CHCl$_3$ | 5a | 25 | 76.5:23.5 | - | - |
| 20[c] | Λ-(S,S)-1h | $^n$hexane | 5a | 38 | 94.5:5.5 | - | - |
| 21[c] | Λ-(S,S)-1h | Et$_2$O | 5a | 25 | 95:5 | - | - |
| 22[c] | Λ-(S,S)-1h | CH$_2$Cl$_2$ | 5a | 29 | 69.5:30.5 | - | - |
| 23[d] | Λ-(S,S)-1h | toluene | TMSN$_3$ | - | - | 76 | 94.5:5.5 |
| 24[d] | Λ-(S,S)-1h | toluene | TsN$_3$ | - | - | trace | N.D. |
| 25[d] | Λ-(S,S)-1h | toluene | NaN$_3$ | - | - | trace | N.D. |

**Table 1 (continued) | Optimization studies for asymmetric Ugi and Ugi-azide reactions**

1a: R' = R'' = $^t$Bu, R''' = $^i$Pr, M = H;
1b: R' = R'' = $^t$Bu, R''' = $^i$Pr, M = Na;
1c: R' = R'' = $^t$Amyl, R''' = $^i$Pr, M = H;
1d: R' = R'' = $^t$Amyl, R''' = $^i$Pr, M = Na;
1e: R' = R'' = $^t$Amyl, R''' = $^t$Bu, M = H;
1f: R' = R'' = $^t$Amyl, R''' = $^t$Bu, M = Na;
1g: R' = R'' = R''' = $^t$Bu, M = H;
1h: R' = R'' = R''' = $^t$Bu, M = Na;

1i: R' = R'' = $^t$Bu, R''' = Bn, M = Na;
1j: R' = R'' = $^t$Bu, R''' = $^t$Bu, M = Na;
1k: R' = R'' = $^t$Bu, R''' = $^s$Bu, M = Na;
1l: R' = R'' = $^t$Bu, R''' = Cy, M = Na;
1m: R' = R'' = R''' = $^t$Bu, M = Li;
1n: R' = R'' = R''' = $^t$Bu, M = K;
1o: R' = R'' = $^t$Bu, R''' = TMS, M = Na;
1p: R' = R'' = $^t$Bu, R''' = TES, M = Na.

| Entry | 1 | solvent | terminating reagents | 6 yield (%)[a] | 6 e.r.[b] | 7 yield (%)[a] | 7 e.r.[b] |
|---|---|---|---|---|---|---|---|
| 26[d] | Λ-(S,S)-1h | toluene | NaN$_3$ + MeNH$_3$$^+$Cl$^-$ | -[f] | - | 79 | 93:7 |
| 27[d] | Λ-(S,S)-1h | toluene | NaN$_3$ + 5b | -[f] | - | 65 | 95:5 |
| 28[e] | Λ-(S,S)-1h | toluene | NaN$_3$ + 5b | -[f] | - | 82 | 95.5:4.5 |

Asymmetric Ugi-4CRs were performed by using 2a (0.12 mmol), 3a (0.10 mmol), 4a (0.12 mmol), 5a (0.10 mmol), 4 Å MS (100 mg) and 1 (0.01 mmol) in toluene (2.0 mL) at −20 °C for 48 h.
[a]Isolated yields were based on 3a.
[b]e.r. values were determined by chiral stationary HPLC.
[c]2a (0.15 mmol), 4a (0.30 mmol), and 5a (0.50 mmol) were employed at −40 °C for 24 h.
[d]Asymmetric Ugi-azide reactions were performed by using 2a (0.15 mmol), 3a (0.10 mmol), 4a (0.30 mmol), terminating reagents (0.30 mmol), 4 Å MS (100 mg) and Λ-(S,S)-1 (0.01 mmol) in toluene (2.0 mL) at −20 °C for 48 h.
[e]NaN$_3$ (0.30 mmol) and 5b (0.40 mmol) were employed at −30 °C.
[f]A little amount of Ugi-4CR product (11% and 13% yield, respectively) was formed. N.D. = not detected.

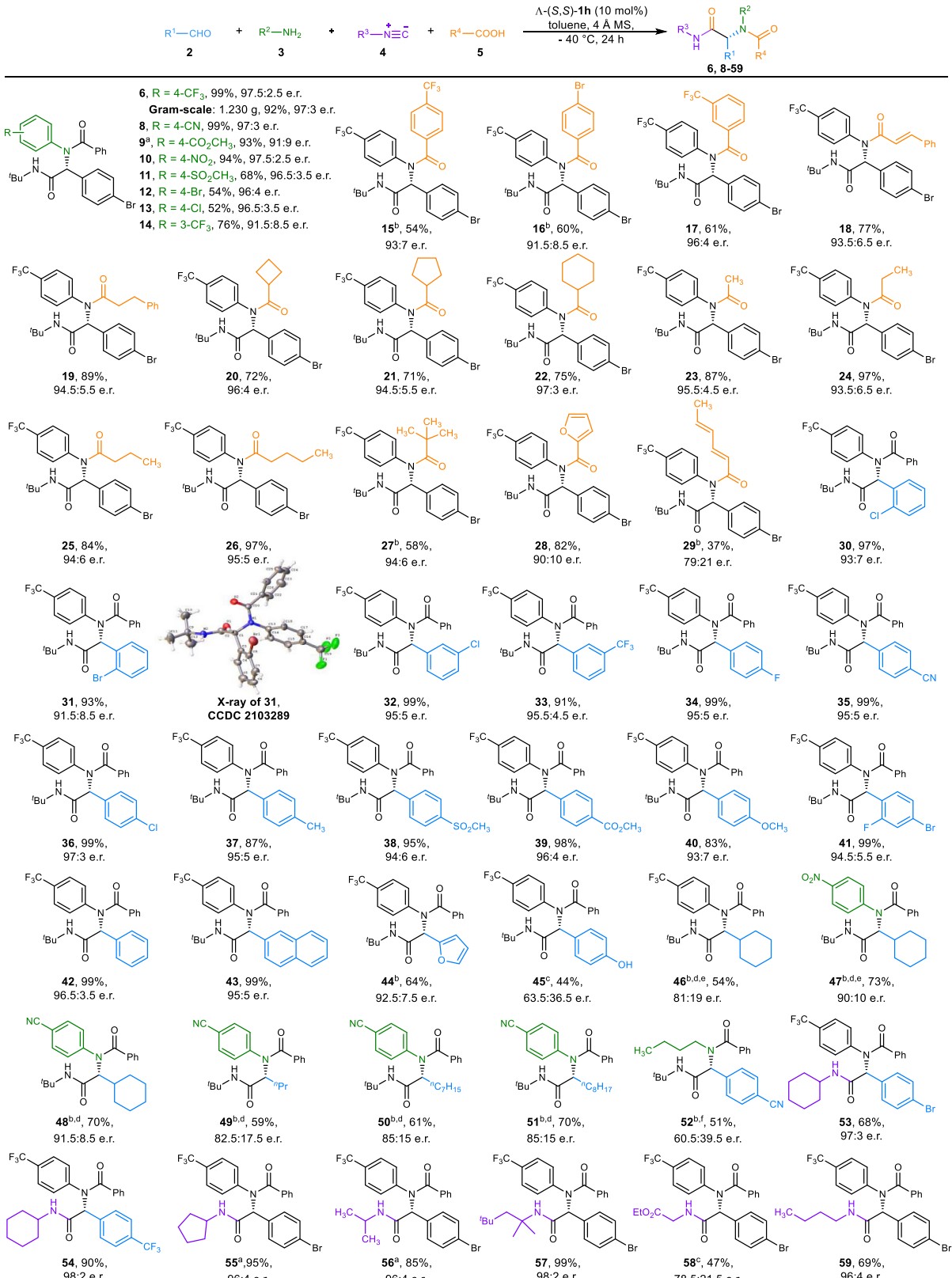

**Fig. 2 | Substrate scope of enantioselective Ugi-4CRs.** All reactions were performed by using **2** (0.15 mmol), **3** (0.10 mmol), **4** (0.30 mmol), **5** (0.50 mmol), 4 Å MS (100 mg) and Λ-(S,S)·**1h** (0.01 mmol) in toluene (2.0 mL) at −40 °C unless otherwise noted; isolated yields were based on amines **3**; e.r. values were determined by chiral stationary HPLC. [a]Run at −30 °C. [b]Run at −20 °C for 36 h. [c]Run at room temperature for 12 h. [d]MeOH (0.10 mL) was added. [e]The ratio of **2/3/4/5** was 1.1/1/3/1. [f]CH₂Cl₂ (2.0 mL) was used instead of toluene.

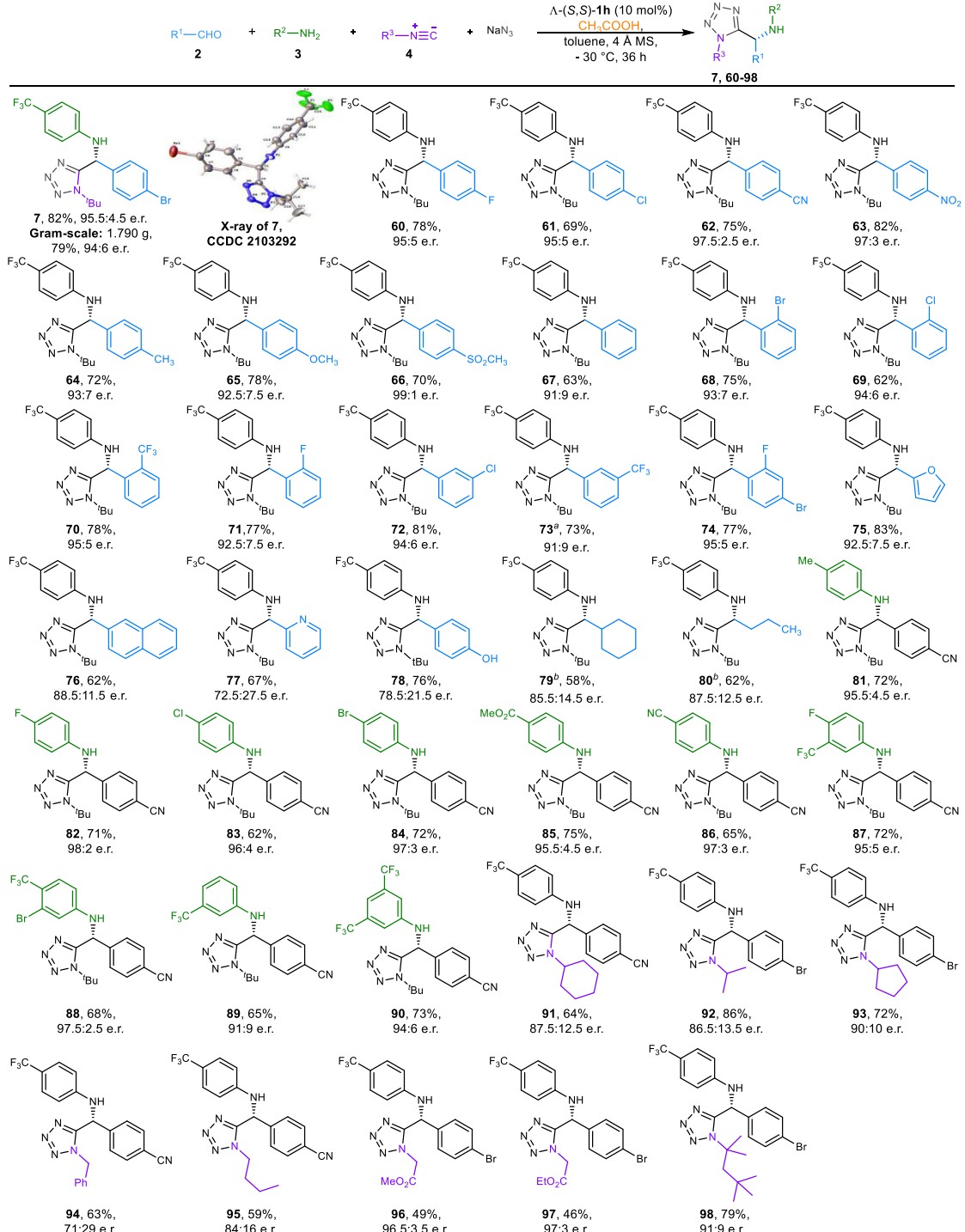

**Fig. 3 | Substrate scope of enantioselective Ugi-azide reactions.** All reactions were performed by using **2** (0.15 mmol), **3** (0.10 mmol), **4** (0.30 mmol), NaN₃ (0.30 mmol), **5b** (0.40 mmol), 4 Å MS (100 mg) and Λ-(S,S)-**1h** (0.01 mmol) in toluene (2.0 mL) at −30 °C unless otherwise noted; isolated yields were based on amines **3**; e.r. values were determined by chiral stationary HPLC. [a]Run at −20 °C. [b]MeOH (0.20 mL) was added.

stereocontrol during the nucleophilic addition of isocyanides to iminiums[31]. Thus, the partial reaction orders of each component and catalyst in both Ugi-4CR and Ugi-azide reactions were determined using the initial rates method (see Supplementary Fig. 2-16 for details)[51–54]. The Ugi-4CR appeared to have a nearly first-order dependence on the anionic stereogenic-at-cobalt(III) complexes **1 h**, aldimine **99**, isocyanide **4**, and acetic acid **5b**, while the Ugi-azide reaction have a first-order dependence on acetic acid **5b** and a zero-order dependence on NaN₃. In particular, the first-order dependence on acetic acid **5b**

indicates that carboxylic acid is involved in rate-limiting steps in both Ugi-4CR and Ugi-azide reactions. It is believed that these experimental evidence of the asymmetric Ugi-azide reaction provide strong support for studying the mechanism of the Ugi-4CR.

Based on the above control experiments, and the previous reports[16–18,31,39,55], a possible reaction mechanism along with transition states was proposed to explain the reaction processes. As shown in Fig. 6, imine **99** was afforded via the condensation of an aldehyde with an amine, which was subsequently activated by anionic stereogenic-at-

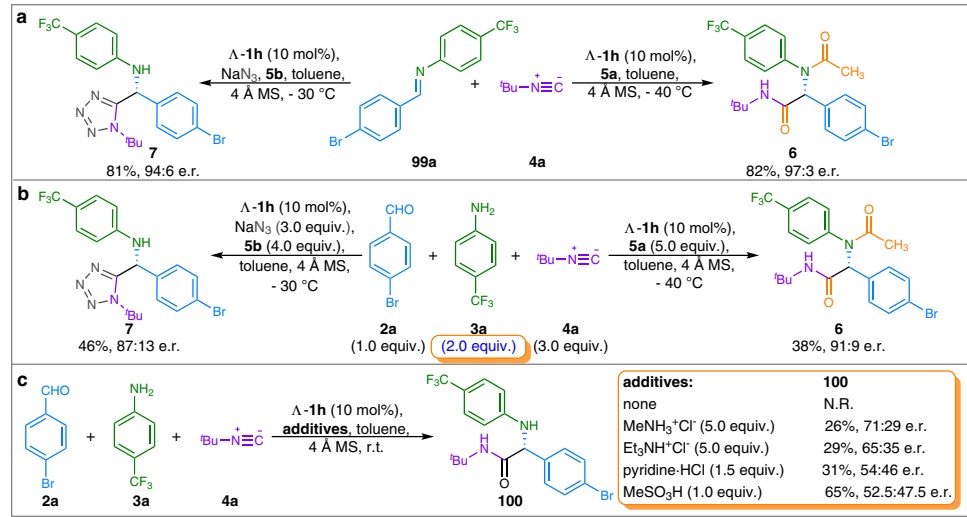

**Fig. 4 | Control experiment. a** Asymmetric Ugi four-component and Ugi-azide reactions with preformed aldimine **99a**. **b** Asymmetric Ugi four-component and Ugi-azide reactions with excess aniline **3a**. **c** Studies of the acid effect on Ugi three-component reactions.

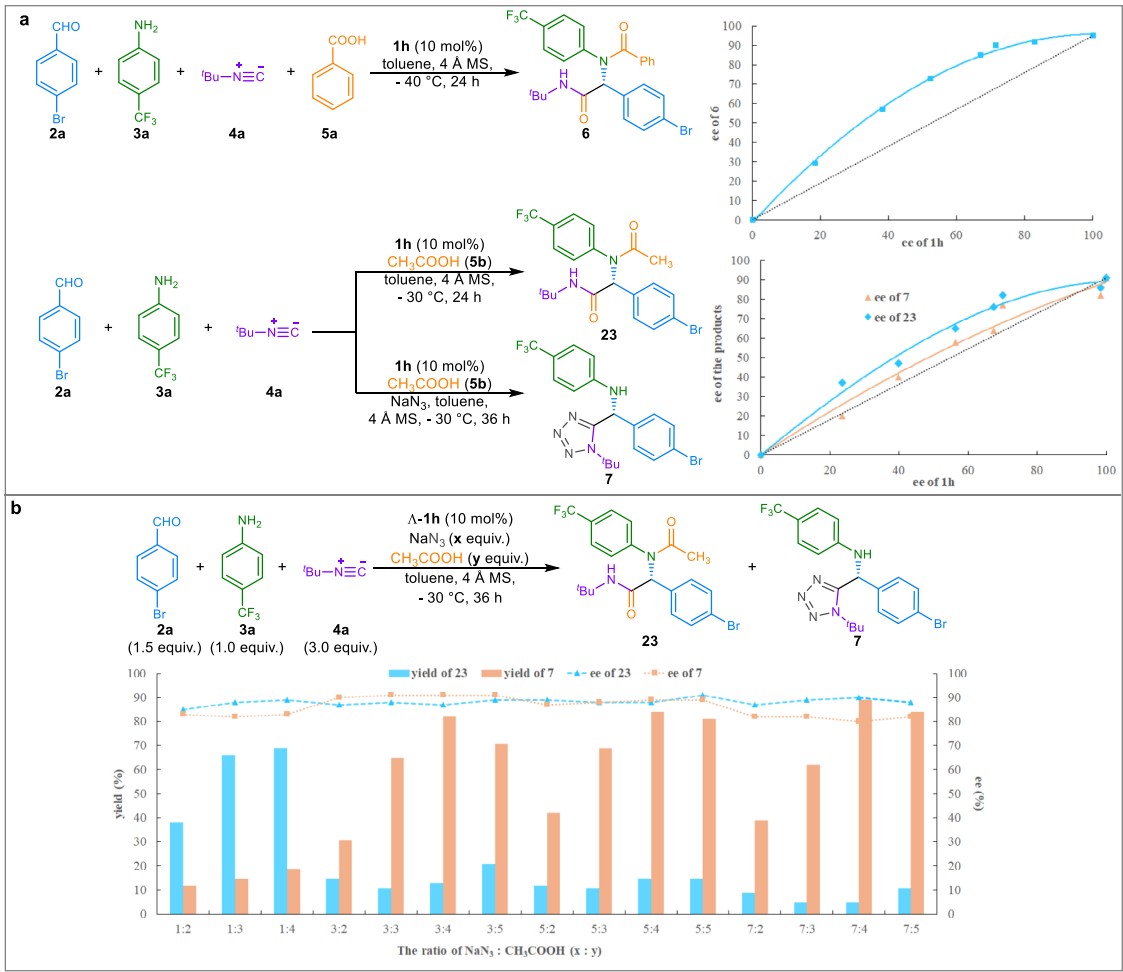

**Fig. 5 | Mechanistic studies. a** The nonlinear effect studies of asymmetric Ugi four-component and Ugi-azide reactions. **b** Comparison of the ratio of NaN$_3$:CH$_3$COOH in the Ugi-azide reactions.

cobalt(III) complexes in the presence of excess carboxylic acids, leading to chiral ion-pair **101**. The activated intermediate **101**, stabilized via the hydrogen-bonding interaction by the carboxylic acid[16], was then subjected to nucleophilic attack by the isocyanide. Since the *Si*-face of the carboxylic acid-activated iminium was shielded by the

bulky tert-butyl substituents of the Schiff base (**TS-II**), isocyanide attacked the *Re*-face of the iminium (**TS-I**) to form the nitrilium intermediate **102** accompanying with carboxylate anion[16]. The key nitrilium intermediate **102** could undergo two different ways by modulating reaction conditions. One is the Ugi-4CR process, in which the

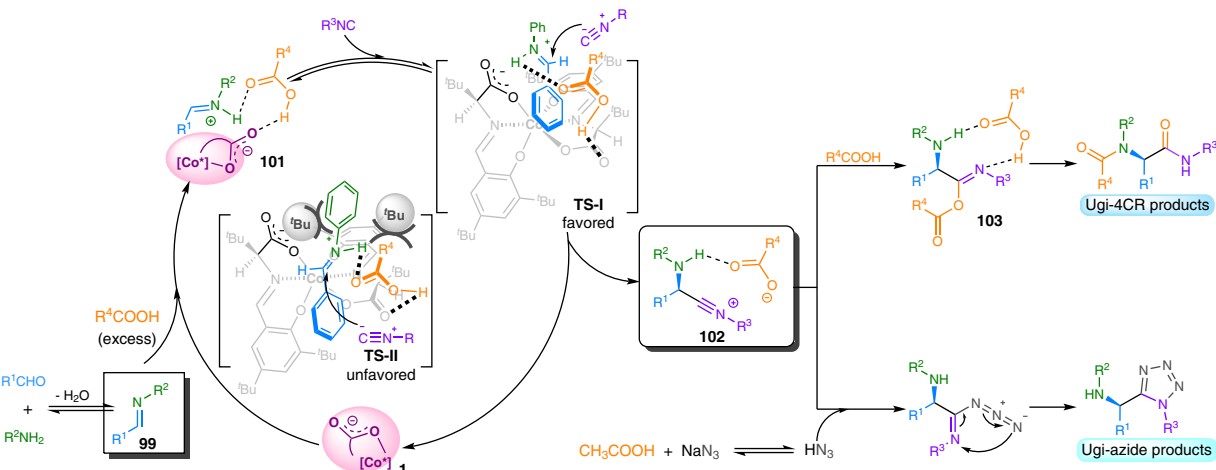

**Fig. 6 | Proposed reaction mechanism.** A proposed catalytic cycle and transition states for the anionic stereogenic-at-cobalt(III) complexes catalysis.

intermediate **102** was trapped by carboxylic acids **5** to afford the imidate **103**, which subsequently underwent Mumm rearrangement to give the desired α-acylamino amides. The other Ugi-azide reaction process started from intermediate **102**, trapped by in situ-formed HN₃, followed by the 1,5-dipolar electrocyclization leads to the formation of tetrazoles.

## Discussion

In summary, inspired by unique privilege of anionic stereogenic-at-cobalt(III) complexes, enantioselective Ugi four-component and Ugi-azide reactions are easy to conduct with mild conditions, leading to highly valuable chiral α-acylamino amides and α-aminotetrazoles. On one hand, we establish the asymmetric Ugi four-component reactions of aldehydes, amines, carboxylic acids and isocyanides in the presence of sodium salt of anionic stereogenic-at-cobalt(III) complexes, providing enantioenriched α-acylamino amides in good to excellent results (up to 99% yield, 98:2 e.r.). On the other hand, with modulating terminating reagents, the chiral transient nitriliums generated from the nucleophilic addition of isocyanides to the chiral ion-pair could be employed as the same key intermediates in the enantioselective Ugi-azide reactions using the same catalyst, leading to chiral α-aminotetrazole derivatives with good to excellent outcomes (up to 83% yield, 99:1 e.r.). These findings not only show the potential of anionic stereogenic-at-cobalt(III) complexes in asymmetric catalysis, but will be able to provide distinct strategies for other isocyanide-based multicomponent reactions.

## Methods

### Materials

Unless otherwise noted, materials were purchased from commercial suppliers and used without further purification. All the solvents were treated according to general methods. Flash column chromatography was performed using 200–300 mesh silica gel. See Supplementary Methods for experimental details.

### Procedure for asymmetric Ugi reactions

A 10-mL oven-dried tube was charged with aldehyde **2** (0.15 mmol), amine **3** (0.10 mmol), catalyst Λ-(S,S)-**1h** (0.01 mmol), 4 Å molecular sieves (100 mg), and toluene (2.0 mL) at room temperature and stirred for 30 min. Then carboxylic acid **5** (0.50 mmol) was added in one portion. The mixture was cooled to −40 °C and stirred for another 30 min. The isocyanide **4** (0.30 mmol) was then added in one portion and the resulting solution was stirred vigorously for 24 h. The reaction was quenched with pre-cooled NEt₃ (−40 °C, 1.0 mmol). The mixture was purified by flash column

chromatography (silica gel, petroleum ether/EtOAc/CH₂Cl₂ = 6:1:1) to give the enantioenriched α-acylamino amide.

### Procedure for asymmetric Ugi-azide reactions

A 10-mL oven-dried tube was charged with aldehyde **2** (0.15 mmol), amine **3** (0.10 mmol), catalyst Λ-(S,S)-**1h** (0.01 mmol), NaN₃ (0.30 mmol), 4 Å molecular sieves (100 mg), and toluene (2.0 mL) at room temperature and stirred for 30 min. Then acetic acid **5b** (0.40 mmol) was added in one portion. The mixture was cooled to −30 °C and stirred for another 30 min. The isocyanide **4** (0.30 mmol) was then added in one portion and the resulting solution was stirred vigorously for 36 h. The reaction was quenched with pre-cooled NEt₃ (−30 °C, 1.0 mmol). The mixture was purified by flash column chromatography (silica gel, petroleum ether/EtOAc/CH₂Cl₂ = 6:1:1) to give the enantioenriched α-aminotetrazole.

### Reporting summary

Further information on research design is available in the Nature Portfolio Reporting Summary linked to this article.

## Data availability

The authors declare that the data supporting the findings of this study are available within the article and its Supplementary Information file. For the experimental procedures, and data of NMR and HPLC analysis, see Supplementary Methods and Charts in Supplementary Information file. The X-ray crystallographic coordinates for structures reported in this article have been deposited at the Cambridge Crystallographic Data Center (**7**: CCDC 2103292, **31**: CCDC 2103289). These data could be obtained free of charge from The Cambridge Crystallographic Data Center via www.ccdc.cam.ac.uk/data_request/cif.

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

## Acknowledgements
We are grateful for financial support from the National Natural Science Foundation of China (grant No. 92156022), Anhui Provincial Natural Science Funds (grant No. 1908085J07 and 1908085QB79), the University Synergy Innovation Program of Anhui Province (grant No. GXXT –2021-059) and Shen-Nong Scholar Program of Anhui Agricultural University.

## Author contributions

J.Y. and H.-Q.C. conceived and directed the project. B.B.S. performed asymmetric Ugi-4CR experiments, and K.L. performed asymmetric Ugi-azide reaction experiments and control experiments. Q.G. performed biological experiments. B.B.S., K.L., and Q.G. contributed equally to this work. W.F., S.L., C.-R.W., and C.-Z.Y. helped with the collection of new compounds and data analysis. J.Y. wrote the paper with input from all other authors. All authors discussed the results and commented on the manuscript.

## Competing interests
The authors declare no competing interests.
