## [Peer Review File · Nature Communications]

Enantioselective Ugi and Ugi-azide reactions catalyzed by anionic stereogenic-at-cobalt(III) complexesREVIEWER COMMENTS

Reviewer #1 (Remarks to the Author):

In this manuscript, the authors reported an approach to switchable enantioselective Ugi-4CRs and Ugi-azide reactions employing anionic stereogenic-at-cobalt(III) complexes as catalysts, delivering α -acylamino amides and α -aminotetrazoles in good to excellent enantioselectivities. Given the fact that there are rather few practical methods for enantioselective Ugi-reactions, this work represents a nice addition!

The manuscript is clearly written and the compounds sufficiently characterized. The scope of the protocol is profoundly investigated. Some points should be addressed. What about the influence of the solvent (only toluene was investigated for the optimization)? The stereogenic-at-cobalt(III) complex catalyst makes use of the abundant natural amino acids. What about the unnatural acids, and the ease/price to generate the homochiral complex. Did they try the reactions with the Λ -(R,R)-1 ? The mechanistic explanation is rather speculative. Did the authors consider e.g. DFT-calculations? After these comments have been addressed, the manuscript could be accepted.

Reviewer #2 (Remarks to the Author):

The manuscript NCOMMS-22-18919-T described enantioselective Ugi and Ugi-azide reactions using anionic chiral Co(III) complexes. It is well-written and disclosed complete study on the development of the titles reactions.

As indicated by the authors, several enantioselective Ugi reactions have been previously reported, including the notable contribution of Tan (ref 31). The originality of the present approach is the activation mode as the authors claim the reaction to occur through the formation of a chiral ion pair, in contrast with the hydrogen-bonding catalysis previously described. The use of this type of anionic chiral Co(III) complexes to activate nitrogenated substrates has been described several times (ref 47), even in the context of multicomponent reactions involving imines (ref 42). In the last part of the article, preliminary bioassay results were reported and the prepared compounds show moderate antifungal activities. Overall, the balance seems slightly positive and this manuscript could be published after consideration of the following points:

1. The scope of isocyanides is very restricted as only two different reagents have been used by the authors (tert-butyl- and cyclohexyl isocyanides). However, the purpose of MCRs, and especially Ugi reactions, is the chemical modularity which allows chemical diversity. Therefore, the authors should add at least 2 or 3 examples using other isocyanides to demonstrate the generality of their approach.
2. In Fig. 3, 5b (acetic acid) could not be considered as a reactant as it is not included in the final product. It should be better considered as an additive and should be put on the arrow.
3. In the proposed catalytic cycle (Fig. 4), the authors describe the intervention of a protonated iminium whereas ref 42 mentions an activation of the imine by the Na⁺ ion. Can the authors precise how they determine the activated form of the imine ?

Reviewer #3 (Remarks to the Author):

The authors describe an interesting catalytic enantioselective Ugi reaction employing a Co complex. What I liked about the work:

- > A new catalytic metal complex inducing stereochemistry in the classical Ugi-4CR and the Ugi tetrazole variation.
- > As opposed to other described enantioselective catalysts the synthesis is straightforward, based on simple and available precursors.
- > Gram scalability

What I didn't like about the work:

- > This is not the first high ee inducing catalytic system described for the Ugi reaction: lack of novelty.
- > The authors used only two isocyanides in their stereoselective reactions: tert-butyl and cyclohexyl. This is not enough to show the generality of the stereo indication of the reaction. At least 10 different isocyanides showing some diversity (aliphatic, aromatic, heteroaromatic, small, bulky, basic) should be used, similarly to the other building blocks.
- > The structural diversity of the other building blocks -COOH, -CHO, -NH₂ is also not high and is restricted to some aromatic substituents and simple aliphatic cases without any orthogonal functional groups, eg. -OH, -OR, -COOR, -NR₂, etc.
- > The authors used an atypical aprotic solvent for the Ugi reaction. Ugi reactions are well performed in protic solvents in 99% of the cases. This has major implications in the usage of a diversity of starting materials.
- > The authors ended their manuscript by describing the biological activities of their compounds. This is totally out of place and distracts the readers from the main part of the article, the stereoselectivity.
- > The paper needs revision and has lots of grammatical errors, e.g. 4-fluoroaniline

In summary, based on the overwhelming negative impression I am not recommending the publication in NatComm. Rather I recommend for publication Organic letters or a similar journal, after having taken into account my critique points.

Reply to the comments of Reviewer #1

We appreciate the Referee for the highly favorable comments and many helpful suggestions! We revised the relevant part in the Manuscript and Supplementary information. All questions are answered below:

(1) What about the influence of the solvent (only toluene was investigated for the optimization)?

Answer: In our early experiments, the influence of the solvents for the asymmetric Ugi-4CR and Ugi-azide reaction has been tested. In order to ensure the coherence of the article, we have added some results into Table 1 in the revised manuscript. Detailed optimizations of the reaction conditions were also added into **Supplementary Table 1** and **Supplementary Table 2** in the revised Supplementary Information (also see as shown below). We hope it is satisfying now.

Supplementary Table 1. Optimization of the reaction conditions for asymmetric Ugi-4CRs

Entry	1	solvent	ratio of 2a:3a:4a:5a	temperature (°C)	yield (%) ^a	e.r. ^b
1	Λ -(S,S)-1a	toluene	1.2:1:1.2:1	-20	54	80.5:19.5
2	Λ -(S,S)-1b	toluene	1.2:1:1.2:1	-20	33	85.5:14.5
3	Λ -(S,S)-1c	toluene	1.2:1:1.2:1	-20	63	80.5:19.5
4	Λ -(S,S)-1d	toluene	1.2:1:1.2:1	-20	61	84.5:14.5
5	Λ -(S,S)-1e	toluene	1.2:1:1.2:1	-20	77	88.5:11.5
6	Λ -(S,S)-1f	toluene	1.2:1:1.2:1	-20	64	93:7
7	Λ -(S,S)-1g	toluene	1.2:1:1.2:1	-20	89	92:8
8	Λ -(S,S)-1h	toluene	1.2:1:1.2:1	-20	72	95:5

1a: R' = R'' = ^tBu, R''' = ⁱPr, M = H; **1i:** R' = R'' = ^tBu, R''' = Bn, M = Na;
1b: R' = R'' = ^tBu, R''' = ⁱPr, M = Na; **1j:** R' = R'' = ^tBu, R''' = ⁱBu, M = Na;
1c: R' = R'' = ^tAmyl, R''' = ⁱPr, M = H; **1k:** R' = R'' = ^tBu, R''' = ^sBu, M = Na;
1d: R' = R'' = ^tAmyl, R''' = ⁱPr, M = Na; **1l:** R' = R'' = ^tBu, R''' = Cy, M = Na;
1e: R' = R'' = ^tAmyl, R''' = ^tBu, M = H; **1m:** R' = R'' = R''' = ^tBu, M = Li;
1f: R' = R'' = ^tAmyl, R''' = ^tBu, M = Na; **1n:** R' = R'' = R''' = ^tBu, M = K;
1g: R' = R'' = R''' = ^tBu, M = H; **1o:** R' = R'' = ^tBu, R''' = TMS, M = Na;
1h: R' = R'' = R''' = ^tBu, M = Na; **1p:** R' = R'' = ^tBu, R''' = TES, M = Na.

9	Λ -(S,S)- 1i	toluene	1.2:1:1.2:1	-20	trace	N.D.
10	Λ -(S,S)- 1j	toluene	1.2:1:1.2:1	-20	27	71:29
11	Λ -(S,S)- 1k	toluene	1.2:1:1.2:1	-20	18	69:31
12	Λ -(S,S)- 1l	toluene	1.2:1:1.2:1	-20	39	65.5:34.5
13	Λ -(S,S)- 1m	toluene	1.2:1:1.2:1	-20	70	94.5:5.5
14	Λ -(S,S)- 1n	toluene	1.2:1:1.2:1	-20	83	93.5:6.5
15	Λ -(S,S)- 1o	toluene	1.2:1:1.2:1	-20	96	93:7
16	Λ -(S,S)- 1p	toluene	1.2:1:1.2:1	-20	40	95:5
17	Δ -(S,S)- 1b	toluene	1.2:1:1.2:1	-20	39	28:72
18	Λ -(S,S)- 1h	toluene	1.2:1:1.2:1	25	79	82.5:17.5
19	Λ -(S,S)- 1h	toluene	1.2:1:1.2:1	0	75	92.5:7.5
20	Λ -(S,S)- 1h	toluene	1.5:1:1.2:1	-20	82	95.5:4.5
21	Λ -(S,S)- 1h	toluene	2:1:1.2:1	-20	80	95.5:4.5
22	Λ -(S,S)- 1h	toluene	1.5:1:2:1	-20	84	95.5:4.5
23	Λ -(S,S)- 1h	toluene	1.5:1:3:1	-20	92	95.5:4.5
24	Λ -(S,S)- 1h	toluene	1.5:1:3:3	-20	99	96:4
25	Λ -(S,S)- 1h	toluene	1.5:1:3:5	-20	99	96.5:3.5
26	Λ -(S,S)- 1h	toluene	1.5:1:3:5	-30	99	97:3
27	Λ -(S,S)- 1h	toluene	1.5:1:3:5	-40	99	97.5:2.5
28	Λ -(S,S)- 1h	CHCl ₃	1.5:1:3:5	-40	15	76.5:23.5
29	Λ -(S,S)- 1h	EtOAc	1.5:1:3:5	-40	N.R.	-
30	Λ -(S,S)- 1h	(CH ₂ Cl) ₂	1.5:1:3:5	-40	37	73:27
31	Λ -(S,S)- 1h	MeCN	1.5:1:3:5	-40	N.R.	-
32	Λ -(S,S)- 1h	ⁿ hexane	1.5:1:3:5	-40	38	94.5:5.5
33	Λ -(S,S)- 1h	Et ₂ O	1.5:1:3:5	-40	15	95:5
34	Λ -(S,S)- 1h	CH ₂ Cl ₂	1.5:1:3:5	-40	19	69.5:30.5
35	Λ -(S,S)- 1h	CH ₃ OH	1.5:1:3:5	-40	N.R.	-
36 ^c	Λ -(S,S)- 1h	toluene	1.5:1:3:5	-40	99	97:3
37 ^d	Λ -(S,S)- 1h	toluene	1.5:1:3:5	-40	76	97:3

Asymmetric Ugi-4CRs were performed by using **2a** (0.12 mmol), **3a** (0.10 mmol), **4a** (0.12 mmol), **5a** (0.10 mmol), 4 Å MS (100 mg) and **1** (0.01 mmol) in toluene (2.0 mL) at -20 °C for 48 h. ^a Isolated yields were based on **3a**. ^b e.r. values were determined by chiral stationary HPLC. ^c Λ -(*S,S*)-**1h** (0.005 mmol) was employed. ^d Λ -(*S,S*)-**1h** (0.002 mmol) was employed. N.R. = no reaction.

Supplementary Table 2. Optimization of the reaction conditions for asymmetric Ugi-azide reactions

Entry	1	R-N₃	solvent	acid	temperature (°C)	yield (%)^a	e.r.^b
1	Δ -(S,S)- 1a	TMSN ₃	toluene	-	25	46	80.5:19.5
2	Δ -(S,S)- 1b	TMSN ₃	toluene	-	25	36	91:9
3	Δ -(S,S)- 1d	TMSN ₃	toluene	-	25	16	73.5:26.5
4	Δ -(S,S)- 1f	TMSN ₃	toluene	-	25	76	69.5:30.5
5	Δ -(S,S)- 1g	TMSN ₃	toluene	-	25	68	90.5:9.5
6	Δ -(S,S)- 1i	TMSN ₃	toluene	-	25	65	72:28
7	Δ -(S,S)- 1m	TMSN ₃	toluene	-	25	77	91.5:8.5
8	Δ -(S,S)- 1n	TMSN ₃	toluene	-	25	56	56.5:43.5
9	Δ -(S,S)- 1o	TMSN ₃	toluene	-	25	51	91.5:8.5
10	Δ -(S,S)- 1p	TMSN ₃	toluene	-	25	34	83:17
11	Δ -(S,S)- 1b	TMSN ₃	toluene	-	25	34	29:71
12	Δ -(S,S)- 1h	TMSN ₃	CCl ₄	-	25	18	89.5:10.5
13	Δ -(S,S)- 1h	TMSN ₃	CH ₂ Cl ₂	-	25	22	63.5:36.5
14	Δ -(S,S)- 1h	TMSN ₃	n ^o hexane	-	25	17	89.5:10.5
15	Δ -(S,S)- 1h	TMSN ₃	(CH ₂ Cl) ₂	-	25	39	61:39
16	Δ -(S,S)- 1h	TMSN ₃	toluene	-	-20	76	94.5:5.5
17	Δ -(S,S)- 1h	TsN ₃	toluene	-	-20	trace	-
18	Δ -(S,S)- 1h	NaN ₃	toluene	-	-20	trace	-
19	Δ -(S,S)- 1h	NaN ₃	toluene	MeNH ₃ ⁺ Cl ⁻	-20	79	93:7
20	Δ -(S,S)- 1h	NaN ₃	toluene	5b	-20	65	95:5
21	Δ -(S,S)- 1h	NaN ₃	toluene	Et ₃ NH ⁺ Cl ⁻	-20	71	90.5:9.5
22 ^c	Δ -(S,S)- 1h	NaN ₃	toluene	5b	-30	82	95.5:4.5
23 ^c	Δ -(S,S)- 1h	NaN ₃	toluene	5c	-30	77	95:5

24 ^c	Λ -(S,S)- 1h	NaN ₃	toluene	5d	-30	69	93:7
25 ^c	Λ -(S,S)- 1h	NaN ₃	toluene	5e	-30	65	92:8
26 ^c	Λ -(S,S)- 1h	NaN ₃	toluene	5f	-30	72	82:18
27 ^c	Λ -(S,S)- 1h	NaN ₃	toluene	CF ₃ COOH	-30	- ^d	N.D.
28 ^c	Λ -(S,S)- 1h	NaN ₃	toluene	CF ₃ SO ₃ H	-30	- ^d	N.D.
29 ^c	Λ -(S,S)- 1h	NaN ₃	toluene	TsOH·H ₂ O	-30	- ^d	N.D.

Asymmetric Ugi-azide reactions were performed by using **2a** (0.15 mmol), **3a** (0.10 mmol), **4a** (0.30 mmol), R-N₃ (0.30 mmol), acid (0.30 mmol), 4 Å MS (100 mg) and **1** (0.01 mmol) in toluene (2.0 mL) for 36 h. ^a Isolated yields were based on **3a**. ^b e.r. values were determined by chiral stationary HPLC. ^c acid (0.40 mmol) was employed. ^d A messy reaction was observed. N.D. = not detected.

(2) The stereogenic-at-cobalt(III) complex catalyst makes use of the abundant natural amino acids. What about the unnatural acids, and the ease/price to generate the homochiral complex. Did they try the reactions with the Λ -(*R,R*)-**1**?

Answer: Considering the price, sodium salts of anionic stereogenic-at-cobalt(III) complexes which derived from both natural (such as *L*-valine **1b** & **1d**, *L*-phenylalanine **1i**, *L*-leucine **1j**, *L*-isoleucine **1k**,) and unnatural amino acids (such as *L*-*tert*-leucine **1f**, **1h**, **1m** and **1n**, *L*- α -cyclohexylglycine **1l**) could be synthesized via the same procedure easily.

Procedures for sodium salts of anionic stereogenic-at-cobalt(III) complexes: the reactions were carried out using salicylaldehydes (5.0 mmol), amino acids (5.0 mmol), Co(NO₃)₂·6H₂O (2.7 mmol), and Na₂CO₃ (2.5 mmol) in EtOH (30 mL) (see ref 47 & 49).

Sodium salts of anionic stereogenic-at-cobalt(III) complexes were screened in the Ugi-4CR and it was found that sodium salt Λ -**1h** which derived from 3,5-di-*tert*-butyl-substituted salicylaldehyde and *L*-*tert*-leucine gave the highest enantiomeric ratio of 95:5 among them. Δ -(*S,S*)-**1b**, the diastereomer of Λ -(*S,S*)-**1b** (see ref 47), enabled the reaction to give the *ent*-**6** in 72:28 e.r. (entry 17, Table 1 in manuscript). We have added relevant results into **Table 1** in the

revised Manuscript.

(3) The mechanistic explanation is rather speculative. Did the authors consider e.g. DFT-calculations?

Answer: Thanks for the good question about the mechanistic explanation. We did consider e.g. DFT-calculations. However, due to the complexity of the catalyst, it is too difficult for us to do the molecular modeling study for now. Further studies on DFT calculations of these catalytic systems in Ugi-type reactions will be carried out. We did additional control experiments and kinetic studies, and revised the proposed mechanism in Fig. 5 in the revised manuscript accordingly to make it more reasonable. We hope it is satisfying now.

In Tan's work (*Science* **2018**, *361*, eaas8707), it was proposed that a hydrogen-bonding-network-stabilized complex, consisted of a chiral phosphoric acid, an imine, and a carboxylic acid, directed the nucleophilic addition of the isocyanide to the activated imine, which was identified as the key of success in achieving excellent enantioselectivity. The proposed mechanism has been supported by DFT calculations and experimentally by comparing the ee of carboxylic acids with different pKa values.

In our work, we did a lot of kinetic studies and control experiments. Firstly, different from Tan's work, both the asymmetric Ugi reaction and Ugi-azide reaction of two equivalents of aniline were able to be carried out in the presence of **excess carboxylic acids**, delivering the corresponding products albeit with slight decreases in enantioselectivity. Interestingly, the Ugi-3CR didn't work in the absence of acid component, whereas the introduction of hydrochlorides or methanesulfonic acid as co-additives could lead the formation of Ugi-3CR product with up to 71:29 e.r. (See the revised Fig. 4). It is suggested that the imine might be activated by a proton transfer in the presence of excess carboxylic acids, leading to iminium intermediate, rather than a hydrogen-bonding activated imine (See ref 31, 52 & 53 in the revised manuscript).

Secondly, Ugi-4CR product **23** was also generated when the asymmetric Ugi-azide reaction was carried out in the presence of NaN₃ and acetic acid. Modulation of the ratio of NaN₃/acetic acid did not avoid the formation of **23** (Fig. 4e, see Supplementary Table 3 for details). Aside from acetic acid, the Ugi-azide reactions with various carboxylic acids were

also examined, and it was found that acetic acid gave the best result and both the steric and the electron nature of the carboxylic acids had a significant effect on the asymmetric introduction (see Supplementary Table 2 for details). **I think these evidences for the asymmetric Ugi-azide reaction also provide strong support for studying the mechanism of the Ugi-4CR.** Moreover, a series of carboxylic acids with different pK_a and steric properties led to the Ugi-4CR products with enantioselectivities ranging from 79:21 to 97:3 e.r. It is revealed that the carboxylic acids did participate in the key step of stereocontrol during the nucleophilic addition of isocyanides to iminiums.

Thirdly, besides the nonlinear effect study, the partial reaction orders of each component and catalyst in both Ugi-4CR and Ugi-azide reaction were also determined using the initial rates method (see Supplementary Fig. 4 for details). The Ugi-4CR appeared to have a nearly first-order dependence on the anionic stereogenic-at-cobalt(III) complexes **1h**, aldimine **99**, isocyanide **4**, and acetic acid **5b**, while the Ugi-azide reaction have a first-order dependence on acetic acid **5b** and a zero-order dependence on NaN_3 . Especially, the first-order dependence on acetic acid **5b** strongly indicates that carboxylic acid is involved in rate-limiting steps in both Ugi-4CR and Ugi-azide reaction.

In Fleurat-Lessard's work (ref 16, *J. Org. Chem.* **2012**, *77*, 1361), the most plausible energy profiles for the Ugi reaction in methanol and in toluene have been performed. In this mechanism study, the non-reversibility of the nitrilium formation indicates that it should be possible to control the stereochemical outcome of the process. It thus gives us the opportunities to settle this enantioselective four-component coupling.

Reply to the comments of Reviewer #2

We are very grateful to the detailed comments of Referee for this Manuscript. According with the kind advice, we revised the relevant part in the Manuscript and Supplementary information. All questions are answered below:

- (1) The scope of isocyanides is very restricted as only two different reagents have been used by the authors (tert-butyl- and cyclohexyl isocyanides). However, the purpose of MCRs, and especially Ugi reactions, is the chemical modularity which allows chemical diversity. Therefore, the authors should add at least 2 or 3 examples using other isocyanides to demonstrate the generality of their approach.

Answer: Thanks for the useful advices from the Referee. We have tried our best to expand the scope of isocyanides in both Ugi-4CRs and Ugi-azide reactions. Although ethyl isocyanoacetate formed **58** with moderate optical purity in Ugi-4CR, primary, secondary, and tertiary alkyl isonitriles formed Ugi products (**53–57**, **59**) in excellent enantioselectivities (up to 98:2 e.r.), regardless of the effect of steric hindrance (see as shown below).

The tolerance of isocyanides in Ugi-azide reactions was also examined. Compared with aliphatic isocyanides (**91-95**, **98**), the reaction of α -isocyanoacetate could deliver Ugi-azide

products **96-97** in medium yield (46-49%) and excellent enantioselectivity (up to 97:3 e.r.). Unfortunately, neither methyl isocyanoacetate nor benzyl isocyanide can undergo the Ugi-4CRs reactions smoothly, while both the Ugi-4CR and Ugi-azide reaction of the 4-methoxyphenyl isocyanide didn't work. We have added relevant results into Fig.2 & Fig. 3 in the revised manuscript. We hope it is satisfying now.

(2) In Fig. 3, 5b (acetic acid) could not be considered as a reactant as it is not included in the final product. It should be better considered as an additive and should be put on the arrow.

Answer: We have corrected this careless error in the revised Manuscript.

(3) In the proposed catalytic cycle (Fig. 4), the authors describe the intervention of a protonated iminium whereas ref 42 mentions an activation of the imine by the Na⁺ ion. Can the authors precise how they determine the activated form of the imine?

Answer: Thanks for the good question about the transition states in Fig.4. In our previous work, the asymmetric Povarov reaction of N-phenylbenzalimine with 2,3-dihydrofuran, which is one of the pericyclic reactions, was carried out **in the absence** of either Brønsted acids or Lewis acids. So, the sodium cation in combination with the weakly coordinating chiral anion was considered as a Lewis acid to activate the 2-azadiene, leading to a possible transition state **TS-I**, as suggested by preliminary DFT calculations (see ref 42). The **Cplx-endo**, leading to the experimentally observed (*S,S*)-enantiomer of major product, was predicted to be more stable energetically than **Cplx-exo** about 1.3 kcal/mol.

In Tan's work (*Science* **2018**, 361, eaas8707), it was proposed that a hydrogen-bonding-

network-stabilized complex, consisted of a chiral phosphoric acid, an imine, and a carboxylic acid, directed the nucleophilic addition of the isocyanide to the activated imine, which was identified as the key of success in achieving excellent enantioselectivity. The proposed mechanism has been supported by DFT calculations and experimentally by comparing the ee of carboxylic acids with different pK_a values.

In our work, we did a lot of kinetic studies and control experiments. Firstly, different from Tan's work, both the asymmetric Ugi reaction and Ugi-azide reaction of two equivalents of aniline were able to be carried out in the presence of **excess carboxylic acids**, delivering the corresponding products albeit with slight decreases in enantioselectivity. Interestingly, the Ugi-3CR didn't work in the absence of acid component, whereas the introduction of hydrochlorides or methanesulfonic acid as co-additives could lead the formation of Ugi-3CR product with up to 71:29 e.r. (See the revised Fig. 4). It is suggested that the imine might be activated by a proton transfer in the presence of excess carboxylic acids, leading to iminium intermediate, rather than a hydrogen-bonding activated imine (See ref 31, 52 & 53 in the revised manuscript).

Secondly, Ugi-4CR product **23** was also generated when the asymmetric Ugi-azide reaction was carried out in the presence of NaN₃ and acetic acid. Modulation of the ratio of NaN₃/acetic acid did not avoid the formation of **23** (Fig. 4e, see Supplementary Table 3 for details). Aside from acetic acid, the Ugi-azide reactions with various carboxylic acids were also examined, and it was found that acetic acid gave the best result and both the steric and the electron nature of the carboxylic acids had a significant effect on the asymmetric introduction (see Supplementary Table 2 for details). **I think these evidences for the asymmetric Ugi-azide reaction also provide strong support for studying the mechanism of the Ugi-4CR.** Moreover, a series of carboxylic acids with different pK_a and steric properties led to the Ugi-4CR products with enantioselectivities ranging from 79:21 to 97:3 e.r. It is revealed that the carboxylic acids did participate in the key step of stereocontrol during the nucleophilic addition of isocyanides to iminiums.

Thirdly, besides the nonlinear effect study, the partial reaction orders of each component and catalyst in both Ugi-4CR and Ugi-azide reaction were also determined using the initial rates method (see Supplementary Fig. 4 for details). The Ugi-4CR appeared to have a nearly first-order dependence on the anionic stereogenic-at-cobalt(III) complexes **1h**, aldimine **99**,

isocyanide **4**, and acetic acid **5b**, while the Ugi-azide reaction have a first-order dependence on acetic acid **5b** and a zero-order dependence on NaN₃. Especially, the first-order dependence on acetic acid **5b** strongly indicates that carboxylic acid is involved in rate-limiting steps in both Ugi-4CR and Ugi-azide reaction.

In Fleurat-Lessard's work (ref 16, *J. Org. Chem.* **2012**, 77, 1361), the most plausible energy profiles for the Ugi reaction in methanol and in toluene have been performed. In this mechanism study, the non-reversibility of the nitrilium formation indicates that it should be possible to control the stereochemical outcome of the process. It thus gives us the opportunities to settle this enantioselective four-component coupling. Due to the complexity of the catalyst, it is too difficult for us to do the molecular modeling study for now. Further studies on DFT calculations of these catalytic systems in Ugi-type reactions will be carried out. We have revised the proposed mechanism in Fig. 5 in the revised manuscript accordingly to make it more reasonable.

Reply to the comments of Reviewer #3

We appreciate the Referee for the favorable comments.

(1) This is not the first high ee inducing catalytic system described for the Ugi reaction: lack of novelty.

Answer: The Ugi multicomponent reactions and related variations have been proven to be atom and step-efficient strategies for construction of highly valuable peptide-like skeletons, nitrogenous heterocycles, and drug candidates. But the stereochemical control remains the most significant challenge still facing the classical Ugi four-component reaction (Ugi-4CR). The enantioselective Ugi-4CR was unconquered until 2018 in the milestone work of Houk, Tan, and coworkers (*Science* **2018**, *361*, eaas8707), in which a hydrogen-bonding-network-stabilized complex, consisted of a chiral phosphoric acid, an imine, and a carboxylic acid, was proposed to direct the nucleophilic addition of the isocyanide to the activated imine. However, **the creation of structurally diverse range of novel catalytic systems and the discovery of new approaches to accommodate a broader scope of terminating reagents** for asymmetric Ugi-4CRs is **still in high demand but challenging**. Moreover, the asymmetric **variations of Ugi-4CR using isosteres**, such as the **Ugi-azide reaction**, have **NOT** been explored so far.

The most significant findings presented in this paper include:

(1) The anionic stereogenic-at-cobalt(III) complexes have been first been convinced to be **very robust catalysts for the switchable enantioselective** multicomponent Ugi and Ugi-azide reactions. The key nitrilium intermediates, generated through the nucleophilic addition of isocyanide to the chiral ion-pair which consists of stereogenic-at-cobalt(III) complexes counteranion and the protonated iminium, could be trapped by either carboxylic acids or *in situ*-generated hydrazoic acid, delivering α -acylamino amides and α -aminotetrazoles in good to excellent enantioselectivities.

(2) On one hand, the asymmetric Ugi-4CRs of aldehydes, amines, carboxylic acids and isocyanides in the presence of sodium salt of anionic stereogenic-at-cobalt(III) complexes were established, providing enantioenriched α -acylamino amides in good to excellent results (up to 99% yield, 98:2 e.r.). On the other hand, with modulating terminating reagents, the chiral transient nitriliums could be employed as the same key intermediates in the

enantioselective Ugi-azide reactions using the same catalyst, leading to chiral α -aminotetrazole derivatives with good to excellent outcomes (up to 83% yield, 99:1 e.r.).

These findings have never been reported thus far.

(3) Preliminary bioassay results indicated that both **α -acylamino amides and tetrazoles** exhibited **potential antifungal activities**.

(4) The concept of the anionic stereogenic-at-cobalt(III) complexes catalysis provides **distinct strategies for addressing challenging stereoselectivity issues** in isocyanide-based multicomponent reactions and will provide a large toolbox amenable for asymmetric catalysis.

(2) The authors used only two isocyanides in their stereoselective reactions: tert-butyl and cyclohexyl. This is not enough to show the generality of the stereo indication of the reaction. At least 10 different isocyanides showing some diversity (aliphatic, aromatic, heteroaromatic, small, bulky, basic) should be used, similarly to the other building blocks. The structural diversity of the other building blocks -COOH, -CHO, -NH₂ is also not high and is restricted to some aromatic substituents and simple aliphatic cases without any orthogonal functional groups, eg. -OH, -OR, -COOR, -NR₂, etc.

Answer: Thanks for the useful advices from the Referee. We have tried our best to expand the scope of isocyanides in both Ugi-4CRs and Ugi-azide reactions. Although ethyl isocyanoacetate formed **58** with moderate optical purity in Ugi-4CR, primary, secondary, and tertiary alkyl isonitriles formed Ugi products (**53–57**, **59**) in excellent enantioselectivities (up to 98:2 e.r.), regardless of the effect of steric hindrance. The tolerance of isocyanides in Ugi-azide reactions was also examined. Compared with aliphatic isocyanides (**91–95**, **98**), the reaction of α -isocyanoacetate could deliver Ugi-azide products **96–97** in medium yield (46-49%) and excellent enantioselectivity (up to 97:3 e.r.). Unfortunately, neither methyl isocyanoacetate nor benzyl isocyanide can undergo the Ugi-4CRs reactions smoothly, while both the Ugi-4CR and Ugi-azide reaction in the presence of the 4-methoxyphenyl isocyanide didn't work (see as shown below).

Except the isocyanides with some orthogonal functional groups (such as $-\text{COOR}$), we also have tried our best to expand the scope of the other components in both Ugi-4CRs and Ugi-azide reactions. The acid components including 2-furoic acid (**28**) and sorbic acid (**29**) were well tolerated under mild conditions, affording the corresponding Ugi-4CR products in moderate to good enantioselectivities. The use of 4-hydroxybenzaldehyde, which contains an acidic functionality, as the aldehyde component led to a significantly diminished enantioselectivity (**45**). Neither the bromoacetic acid nor the benzyloxyacetaldehyde undergo the asymmetric Ugi-4CR. Tetrazole **78** formed from 4-hydroxybenzaldehyde could be obtained with moderate enantioselectivity. However, the introduction of both benzyloxyacetaldehyde and aliphatic amines cannot lead to the desired products.

We have added relevant results into Fig.2 & Fig. 3 in the revised manuscript. We hope it is satisfying now.

- (3) The authors used an atypical aprotic solvent for the Ugi reaction. Ugi reactions are well performed in protic solvents in 99% of the cases. This has major implications in the usage of a diversity of starting materials.

Answer: As we know, polar protic solvents (e.g., MeOH) are the conventional reaction media for the racemic Ugi-4CR. However, competition from background reactions occur spontaneously in an appropriate solvent at room temperature. The discovery of catalytic asymmetric variants of the Ugi-4CR was hampered by the polar-protic solvent which competes for the catalyst's electrophilic center as well as by the strongly nucleophilic and complexing nature of the amine component (*Angew. Chem. Int. Ed.* **2018**, *57*, 16266; *Acc. Chem. Res.* **2018**, *51*, 1290). Noteworthy, in Tan's work (*Science* **2018**, *361*, eaas8707), the solvents used in these procedures are dichloromethane and cyclohexane, atypical Ugi solvents. In our opinion, on the contrary, the use of atypical aprotic solvent (toluene) for the Ugi-4CR and Ugi-azide reaction has more important implications in the usage of a diversity of starting materials.

- (4) The authors ended their manuscript by describing the biological activities of their compounds. This is totally out of place and distracts the readers from the main part of the article, the stereoselectivity.

Answer: In my opinion, the study of asymmetric synthetic methodologies should not only show the high efficiency and stereoselectivity of the synthetic strategies, but also show the potential application prospects. Both Ugi-4CR and Ugi-azide reactions which are useful for the synthesis of various peptide-like skeletons, heterocyclic compounds, and drug candidates, have found a wide scope of applications in pharmaceutical discovery and production. In our manuscript, these chiral products afforded from classical enantioselective Ugi-4CR and Ugi-azide reactions have shown potential antifungal activities and can be regarded as promising candidates in the search for new pesticide scaffolds. So, we do not think this part of describing the biological activities will distract the readers from the main part of the article.

- (5) The paper needs revision and has lots of grammatical errors, e.g. 4-fluroaniline.

Answer: We have corrected this careless error in the revised Manuscript and checked the whole manuscript carefully. We hope it is satisfying now.

REVIEWERS' COMMENTS

Reviewer #2 (Remarks to the Author):

The manuscript NCOMMS-22-18919A is a revised version of NCOMMS-22-18919-T. All concerns raised in my previous review have been addressed by the authors. Indeed, they have provided additional examples (to demonstrate the generality of the method) and detailed information (to support their hypotheses). Therefore, I support publication of the present version of the manuscript.

Reviewer #3 (Remarks to the Author):

None of my comments/critiques were sufficiently addressed and I am therefore rejecting the paper. In order of descending importance:

1) Enantioselective Ugi reactions were previously described. Therefore, there is a FUNDAMENTAL LACK OF NOVELTY. Now, mere incremental improvements cannot expect to be published anymore in forefront journals.

2) The choice of the authors of isocyanide diversity is not satisfactory. They used unsubstituted, rather simple unpolar substituents. Ugi reaction however has a much broader substituent scope (see also point 4). What about using an isocyanide with a free OH group or a tertiary amine (morpholino ethyl is even commercially available)? It is well known that such substituents can change stereoselectivity considerably (see for example the enantioselective Passerini reaction). If authors do not provide detailed scope and limitations of a newly found reaction condition, here enantioselective catalysis, then future practitioners have to try and find out by themselves real-world examples. A major reason why academic synthetic 'innovations' are not finding applications in the industry is exactly that only simple substituents ('methyl, ethyl, fusil') were exercised but not demanding real-world examples. The authors exclusively used such simple aliphatic and aromatic substituents. In the few other cases, the ee also dropped considerably, such as in the p-hydroxy benzaldehyde. What about OH in m-, or o-position, what about heterocycles such as m-, o-, p-pyridine etc etc. The described examples do not reflect anyhow the diversity of functional groups compatible for Ugi reactions. Why didn't they apply their methodology to a real-world example that they mentioned in the introduction?

3) Ugi reactions run mostly in protic solvent for a good reason: because of the solubility of many starting materials. See my critique point in 2. This is likely why the authors avoided using more protic substituents. Therefore I cannot at all understand the authors statement: 'In our opinion, on the contrary, the use of atypical aprotic solvent (toluene) for the Ugi-4CR and Ugi-azide reaction has more important implications in the usage of a diversity of starting materials.' What does this mean? Where is the advantage of polarity restriction in terms of scope of the reaction?

4) I am still of the opinion that the description of some random bioactivities severely distracts from the main topic of the manuscript the enantioselective Ugi reaction. It is well established in hundreds of previous publications (with a much better mechanism-based focus) that Ugi products lead to bioactive compounds. A Nature-type manuscript should not resemble a generalists vendor's tray offering a little bit of everything, but rather surprise the reader by a specific detailed finding and thereby lead to an advancement of the area.

THIS I AM SORRY IS NOT THE CASE HERE.

Response to the comments of Reviewer #2

The manuscript NCOMMS-22-18919A is a revised version of NCOMMS-22-18919-T. All concerns raised in my previous review have been addressed by the authors. Indeed, they have provided additional examples (to demonstrate the generality of the method) and detailed information (to support their hypotheses). Therefore, I support publication of the present version of the manuscript.

Response: We sincerely thank the Reviewer for the positive evaluation and strong support of this work.

Response to the comments of Reviewer #3

(1) Enantioselective Ugi reactions were previously described. Therefore, there is a FUNDAMENTAL LACK OF NOVELTY. Now, mere incremental improvements cannot expect to be published anymore in forefront journals.

Response: In our work, anionic stereogenic-at-cobalt(III) complexes have been first been convinced to be very robust catalysts for both the enantioselective Ugi four-component and Ugi-azide reactions. In particular, the asymmetric variations of Ugi-4CR using isosteres, such as the Ugi-azide reaction, have NOT been explored so far. It is believed that the concept of the anionic stereogenic-at-cobalt(III) complexes catalysis will provide distinct strategies for addressing challenging stereoselectivity issues in isocyanide-based multicomponent reactions and a useful toolbox amenable for asymmetric catalysis.

(2) The choice of the authors of isocyanide diversity is not satisfactory. They used unsubstituted, rather simple unpolar substituents. Ugi reaction however has a much broader substituent scope (see also point 4). What about using an isocyanide with a free OH group or a tertiary amine (morpholino ethyl is even commercially available)? It is well known that such substituents can change stereoselectivity considerably (see for example the enantioselective Passerini reaction). If authors do not provide detailed scope

and limitations of a newly found reaction condition, here enantioselective catalysis, then future practitioners have to try and find out by themselves real-world examples. A major reason why academic synthetic 'innovations' are not finding applications in the industry is exactly that only simple substituents ('methyl, ethyl, fusil') were exercised but not demanding real-world examples. The authors exclusively used such simple aliphatic and aromatic substituents. In the few other cases, the ee also dropped considerably, such as in the p-hydroxy benzaldehyde. What about OH in m-, or o-position, what about heterocycles such as m-, o-, p-pyridine etc etc. The described examples do not reflect anyhow the diversity of functional groups compatible for Ugi reactions. Why didn't they apply their methodology to a real-world example that they mentioned in the introduction?

Answer: Caused by the Covid-19 pandemic, we have tried our best to expand the scope of isocyanides in both Ugi-4CRs and Ugi-azide reactions. As you can see, ee dropped considerably when the p-hydroxy benzaldehyde was employed. The aliphatic aldehydes underwent Ugi 4-CRs and Ugi-azide reactions to generate the desired products with moderate to good enantioselectivities when methanol was used as a co-solvent. In Ugi-azide reaction, tetrazoles **77** formed from 2-pyridyl carboxaldehyde was obtained with moderate enantioselectivity. These examples showed that some functional groups, such as hydroxyl, pyridyl, and polar-protic solvents have obvious influences on the enantioselectivity. Further studies on the Ugi and post-Ugi reactions of substrates containing diverse functional groups are underway.

(3) Ugi reactions run mostly in protic solvent for a good reason: because of the solubility of many starting materials. See my critique point in 2. This is likely why the authors avoided using more protic substituents. Therefore I cannot at all understand the authors statement: 'In our opinion, on the contrary, the use of atypical aprotic solvent (toluene) for the Ugi-4CR and Ugi-azide reaction has more important implications in the usage of a diversity of starting materials.' What does this mean? Where is the advantage of polarity restriction in terms of scope of the reaction?

Answer: As we know, competition from background reactions occur spontaneously in polar protic solvents (e.g., MeOH) even at -40 °C (*Acc. Chem. Res.* **2018**, *51*, 1290). As we mentioned above, the Ugi-4CRs and Ugi-azide reactions did not occur in toluene at -

40 °C when aliphatic aldehydes were used. However, the aliphatic aldehydes underwent Ugi 4-CRs and Ugi-azide reactions smoothly to generate the desired products when methanol was used as a co-solvent. It should not be attributed to the solubility of starting materials, but to the complexity of the reaction mechanism.

As Prof. Dömling said, ‘Whereas the P-3CR prefers apolar-aprotic solvents such as THF or dichloromethane, the U-4CR runs best in polar-protic solvents. ... In the U-4CR the key intermediate is the Schiff base which is less reactive than the oxo component in the P-3CR and needs activation through protonation by the carboxylic acid or a Lewis acid and is then attacked by the nucleophilic isocyanide in the stereogenic step. The charged complex of the U-4CR is best stabilized in protic-polar solvents. Consequently, chiral Lewis acid induced Passerini reactions were reported early on and now optimized procedures are available. The discovery of catalytic asymmetric variants of the U-4CR was hampered by the polar-protic solvent which competes for the catalyst’s electrophilic center as well as by the strongly nucleophilic and complexing nature of the amine component.’ (*Angew. Chem. Int. Ed.* **2018**, *57*, 16266).

(4) I am still of the opinion that the description of some random bioactivities severely distracts from the main topic of the manuscript the enantioselective Ugi reaction. It is well established in hundreds of previous publications (with a much better mechanism-based focus) that Ugi products lead to bioactive compounds. A Nature-type manuscript should not resemble a generalists vendor's tray offering a little bit of everything, but rather surprise the reader by a specific detailed finding and thereby lead to an advancement of the area.

Answer: Thanks for the advices from the Reviewer, we have revised the manuscript and the bioassay results have been placed in the Supplementary Information.